# Optical circulation in a multimode optomechanical resonator

Freek Ruesink[1], John P. Mathew[1], Mohammad-Ali Miri[2], Andrea Alù [2,3,4,5] & Ewold Verhagen[1]

Breaking the symmetry of electromagnetic wave propagation enables important technological functionality. In particular, circulators are nonreciprocal components that can route photons directionally in classical or quantum photonic circuits and offer prospects for fundamental research on electromagnetic transport. Developing highly efficient circulators thus presents an important challenge, especially to realise compact reconfigurable implementations that do not rely on magnetic fields to break reciprocity. We demonstrate optical circulation utilising radiation pressure interactions in an on-chip multimode optomechanical system. Mechanically mediated optical mode conversion in a silica microtoroid provides a synthetic gauge bias for light, enabling four-port circulation that exploits tailored interference between appropriate light paths. We identify two sideband conditions under which ideal circulation is approached. This allows to experimentally demonstrate ~10 dB isolation and <3 dB insertion loss in all relevant channels. We show the possibility of actively controlling the circulator properties, enabling ideal opportunities for reconfigurable integrated nanophotonic circuits.

[1] Center for Nanophotonics, AMOLF, Science Park 104, 1098 XG Amsterdam, The Netherlands. [2] Department of Electrical and Computer Engineering, The University of Texas at Austin, Austin, TX 78712, USA. [3] Photonics Initiative, Advanced Science Research Center, City University of New York, New York 10031, USA. [4] Physics Program, Graduate Center, City University of New York, New York 10016, USA. [5] Department of Electrical Engineering, City College of The City University of New York, New York 10031, USA. These authors contributed equally: Freek Ruesink, John P. Mathew. Correspondence and requests for materials should be addressed to E.V. (email: verhagen@amolf.nl)

Optical circulators route photons in a unidirectional fashion among different ports, with diverse applications in advanced communication systems, including dense wavelength division multiplexing and bi-directional sensors and amplifiers. Their operation offers opportunities for routing quantum information[1,2] and to realise photonic states whose propagation in a lattice is topologically protected[3,4]. Traditionally, nonreciprocal elements, such as circulators and isolators, have relied on applied magnetic bias fields to break time-reversal symmetry and Lorentz reciprocity. While significant progress has been made towards introducing magneto-optic materials in photonic circuits[5], realising low-loss, linear, efficient, and compact photonic circulators on a chip remains an outstanding challenge. In recent years, coupled-mode systems that create an effective magnetic field using parametric modulations have been recognised as powerful alternatives[1,6–12]. Optomechanical systems, where multiple optical and mechanical modes are coupled through radiation pressure[13–15], provide an effective platform to realise such modulation[16–18]. Nonreciprocal transmission and optical isolation were demonstrated in several optomechanical implementations, including ring resonators, photonic crystal nanobeams, and superconducting circuits[19–23]. Recently, non-reciprocal circulation for microwave signals was reported in a superconducting device exhibiting directional frequency conversion[24].

Here, by leveraging a synthetic gauge field created in a multimode optomechanical system, we realise a compact and highly reconfigurable on-chip circulator in the photonic (tele-communication) domain. With this, we establish a magnet-free circulator at optical wavelengths. The working principle, relying on tailored interfering paths mediated by optical dissipation channels, allows for operation at equal-frequency input and output fields and could be applied in a wide variety of platforms. We reveal the importance of control fields and port couplings to regulate the nonreciprocal response and identify two distinct regimes, with and without employing optomechanical gain, where this response can approach ideal circulation.

## Results

**Optomechanical mode conversion and circulation.** Our experimental system consists of a high-$Q$ microtoroid resonator[25] that is simultaneously side-coupled to two tapered optical fibres, forming four ports through which light can enter and exit (Fig. 1a). The microtoroid supports nearly degenerate odd and even optical modes—superpositions of clockwise and counterclockwise propagating waves—coupled through a mechanical breathing mode, to form a three-mode optomechanical system. A control beam incident through port 1 populates both modes, labelled by $i = \{1, 2\}$, with intracavity control fields $\alpha_i$ that exhibit $\pi/2$ phase difference[26]. When detuned from the cavity, these control fields induce linear couplings between the mechanical resonator and both cavity modes at rates $g_i = g_0\alpha_i$, where $g_0$ is the vacuum optomechanical coupling rate[13]. For red-detuned control (Fig. 1b), photon transfer from mode 1 to mode 2 via the mechanical resonator then takes place at rate $\mu_m = 2g_1^*g_2/\Gamma_m$, with $\Gamma_m$ the mechanical linewidth. In contrast, transfer from mode 2 $\longrightarrow$ 1 occurs at rate $\mu_m^*$, i.e., with opposite phase (see Methods for origin of $\mu_m$). The control field thus biases the mode conversion process with maximally nonreciprocal phase $\Delta\phi \equiv \arg(\mu_m) = \pi/2$[18,20]. This phase is reminiscent of a synthetic d.c. magnetic flux that breaks time-reversal symmetry[27] and can, under appropriate conditions, serve to create an optical circulator. Importantly, such a circulator is linear, up to probe levels that approach the intracavity control field amplitude.

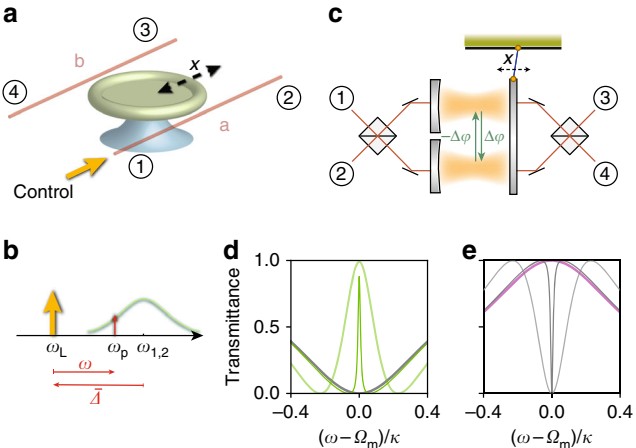

**Fig. 1** Optical nonreciprocity in a multimode optomechanical system. **a** An optomechanical microtoroid coupled to optical fibres a (ports 1, 2) and b (ports 3, 4). Even and odd optical whispering gallery modes interact with a mechanical radial breathing mode through radiation pressure. A control beam, entering in port 1, can induce nonreciprocal transmission in the direct (1 ↔ 2, 3 ↔ 4) and add-drop (2 ↔ 3, 4 ↔ 1) channels. **b** Control ($\omega_L$) and probe ($\omega_p$) frequencies with respect to the cavity frequencies $\omega_{1,2}$. **c** An analogous Fabry–Pérot model for the three-mode system is shown. The optical mode conversion mediated by the mechanical mode imprints a nonreciprocal phase $\Delta\phi$ on forward and reverse intercavity mode transfers. **d**, **e** Theoretically predicted probe transmittances for the **d** direct ($|s_{21}|^2$: green shaded lines, $|s_{12}|^2$: grey shaded line) and add-drop channels ($|s_{32}|^2$: purple shaded line, $|s_{23}|^2$: grey shaded lines) for $\bar{\Delta} = -\Omega_m$, negligible intrinsic losses and two different cooperativities ($\mathcal{C} = 15$ and 200). The nonreciprocal response is inferred from $s_{ij} \neq s_{ji}$. Its strength and bandwidth are increased through higher cooperativity (lines of lighter shade). The vertical axis is the same for **d**, **e**

Our system is fully equivalent to the general two-cavity Fabry–Pérot model sketched in Fig. 1c, where the beamsplitters impart a $\pi/2$ phase shift on reflections from either side. From each port the two cavities are then excited with the same $\pi/2$ phase difference as the even and odd modes in the ring resonator. In absence of optomechanical coupling, this realises a reciprocal add-drop filter: off-resonant light propagates from port 1(3) to port 2(4) and vice versa, while resonant light is routed from 1(2) to 4(3). However, in the presence of a control beam at a frequency $\omega_L$ detuned from the cavity frequencies $\omega_{1,2}$ by $\bar{\Delta} = -\Omega_m$, with $\Omega_m$ the mechanical resonance frequency, a weaker probe beam at frequency $\omega_p = \omega_L + \omega$ (Fig. 1b) experiences an optomechanically induced transparency (OMIT[28]) window when $\omega \approx \Omega_m$: the probe is routed from port 1 to 2 rather than dropped in port 4. This behaviour is quantified in Fig. 1d, e, which shows scattering matrix elements $s_{ij}$, signalling transmission from port $j$ to $i$, for two control powers. These are obtained from a rigorous coupled-mode model describing the optomechanical three-mode system (see Methods). The response is clearly nonreciprocal: the OMIT window that is observed for $s_{21}$ (green curves, Fig. 1d) does not occur for probe light incident from port 2 (grey), which remains dropped in port 3 (purple curve, Fig. 1e). Likewise, light from port 3 is suppressed (grey) and routed to port 4, etc. The optomechanical interactions in this multimode system thus allow nonreciprocal add-drop functionality, which can also be related to momentum-matching of intracavity control and probe fields[16]. For proper bias conditions, this yields a circulator, i.e., a control beam incident from port 1 or 3 allows probe transmission 1 $\longrightarrow$ 2, 2 $\longrightarrow$ 3, 3 $\longrightarrow$ 4, 4 $\longrightarrow$ 1, whereas a control from port 2 or 4 reverses the circulation direction.

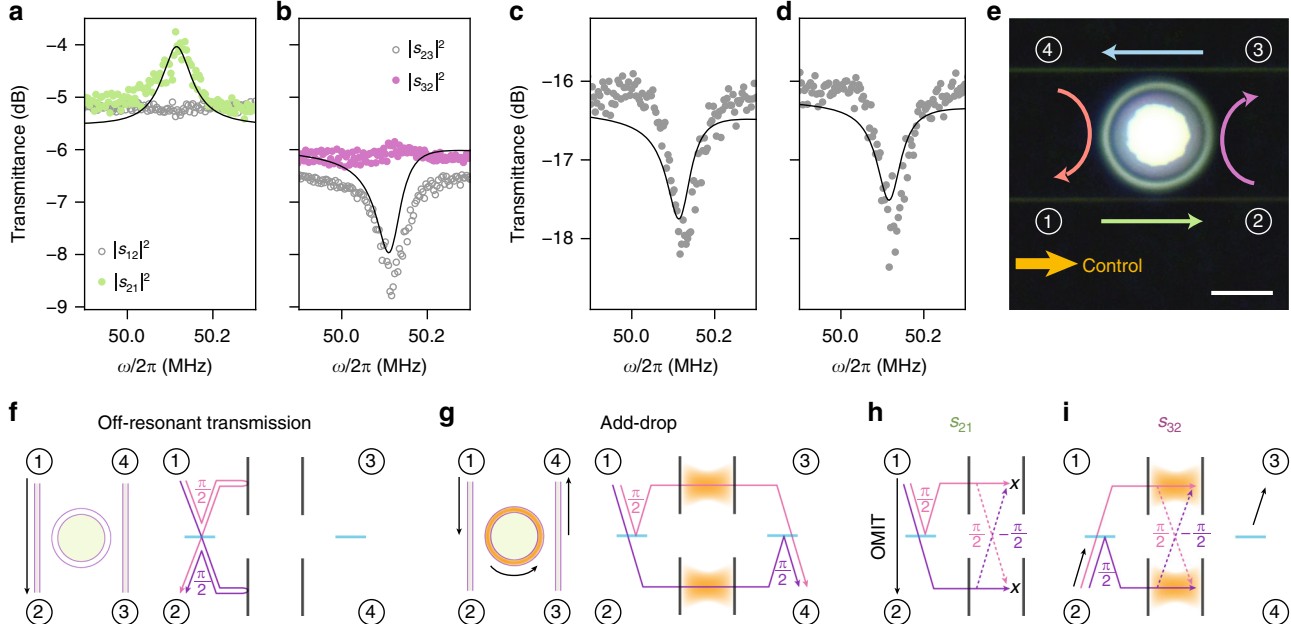

**Fig. 2** Nonreciprocal mode transfer and circulation. **a, b** Measured probe transmittance as a function of probe-control detuning ($\omega = \omega_p - \omega_L$) for the **a** direct and **b** add-drop channels when $\overline{\Delta}_{1,2} \approx -\Omega_m$. When probe and control beams co-propagate in the $1 \leftrightarrow 2$ channel, nonreciprocal optomechanically induced transparency (OMIT) is observed in both channels. Transmittance in directions $1 \longrightarrow 2$ (green data points) and $2 \longrightarrow 3$ (purple data points) is larger than their reverse (grey data points), yielding circulation. The vertical axis is the same for **a, b. c, d** A small optical nondegeneracy yields non-zero **c** reflection ($|s_{33}|^2$: grey coloured data) and **d** cross-coupling ($|s_{31}|^2$: grey coloured data), suppressed by the optomechanical interaction. The vertical axis is the same for **c, d**. The black lines in **a–d** are theoretical fits for $|s_{ij}|^2$ using a global fit to all 16 transmittance spectra (see Supplementary Note 2 and Supplementary Fig. 2). **e** Microscope image of the microtoroid and the tapered fibres (scalebar corresponds to 20 μm), indicating ports and circulation direction. **f–g** Phase pickup and corresponding constructively interfering light paths (pink and purple solid lines) in an add-drop filter without control beam. **f** Direct transmission from $1 \longrightarrow 2$ when the probe does not excite the optical modes. **g** Add-drop functionality from $1 \longrightarrow 4$ when the probe excites both modes with $\pi/2$ phase difference. **h, i** With control beam, nonreciprocal add-drop operation is achieved using the mechanically mediated mode transfer paths (pink and purple dotted lines), whose opposite phases cause **h** destructive interference in the cavities for light entering port 1 (thus, OMIT from $1 \longrightarrow 2$), and **i** constructive interference in the cavities for light from port 2 (giving transmission $2 \longrightarrow 3$). For **f–i**, the black arrows denote overall path of transmitted light and orange shading denotes cavity modes that are excited

In experiment, we measure circulation using a heterodyne technique (see Methods) to obtain transmission spectra for all 16 port combinations that constitute the $4 \times 4$ scattering matrix. The studied microtoroid has nearly degenerate optical modes at $\omega_i/2\pi = 196.7$ THz with intrinsic loss rate $\kappa_0/2\pi = 8.3$ MHz, and supports a mechanical breathing mode at $\Omega_m/2\pi = 50.12$ MHz ($\Gamma_m/2\pi = 62$ kHz). The two optical modes are assumed to have equal total loss rates $\kappa = \kappa_0 + \kappa_a + \kappa_b$, where $\kappa_{a,b}$ are exchange losses to waveguides a and b. Figure 2a, b shows the measured probe transmittances in the direct ($1 \leftrightarrow 2$) and add-drop ($2 \leftrightarrow 3$) channels when the control is detuned to the lower mechanical sideband of the cavity $\left(\overline{\Delta}_{1,2} \approx -\Omega_m\right)$. OMIT is observed in the co-propagating direct channel $\left(|s_{21}|^2\right)$, accompanied by reduced transmission in the add-drop channel $\left(|s_{23}|^2\right)$. The OMIT features are absent in reverse operation of the device, which thus exhibits nonreciprocal transport. The overall effect is optical circulation in the direction $1 \longrightarrow 2 \longrightarrow 3 \longrightarrow 4 \longrightarrow 1$ (see Supplementary Fig. 2 for all 16 scattering matrix elements).

**Role of port couplings and interference.** To understand the role of mechanically mediated mode transfer in the observed response, we recapitulate the operation of conventional add-drop filters: when a probe signal from port 1(3) does not excite the two modes (e.g., because it is off-resonance), it interferes constructively in port 2(4) and vice versa (Fig. 2f). When the probe does excite both modes, their $\pi/2$ phase difference means that light constructively interferes in the drop port (e.g., $1 \longrightarrow 4$, Fig. 2g). However, a control beam reconfigures this behaviour through

nonreciprocal mode conversion. Now a probe signal incident from port 1 reaches each mode either directly or via the parametric mode conversion process. As a result of the $\pm\pi/2$ phase shift associated with the mode transfer paths, the direct and mode transfer path destructively interfere in both cavities (Fig. 2h). Consequently, a probe from port 1 does not excite the modes and is transmitted to port 2 instead of 4. In contrast, a probe from port 2 excites both modes with opposite phase difference, such that interference inside the cavities is constructive and light is routed to port 3 (Fig. 2i). Therefore, circulation can only occur in a properly configured two-mode optical system. In particular, the presence of a direct scattering path between ports 1 and 2 (and $3 \leftrightarrow 4$) is crucial to induce the appropriate phase shift and interference causing circulation.

We thus recognise the importance of the way the ports are coupled to the optical modes, and to each other: to create a circulator with only two optical modes, the employed passive optical system is configured such that light passes from one port to a second when not interacting with the cavity modes, and to a third port when the cavities are excited. That port takes the role of intrinsic cavity loss that dissipates energy in the backward direction when such a system is employed as an isolator[18]. In comparison, for multimode optomechanical isolators designed with end-coupled ports[21–24], the mechanical bath that provides necessary dissipation cannot be simply replaced by one or two additional optical output ports to create circulation. Instead, it requires the addition of extra cavity modes and multiple control fields[23,24]. However, we show here that this can be avoided by carefully choosing the port coupling conditions, enabling four-

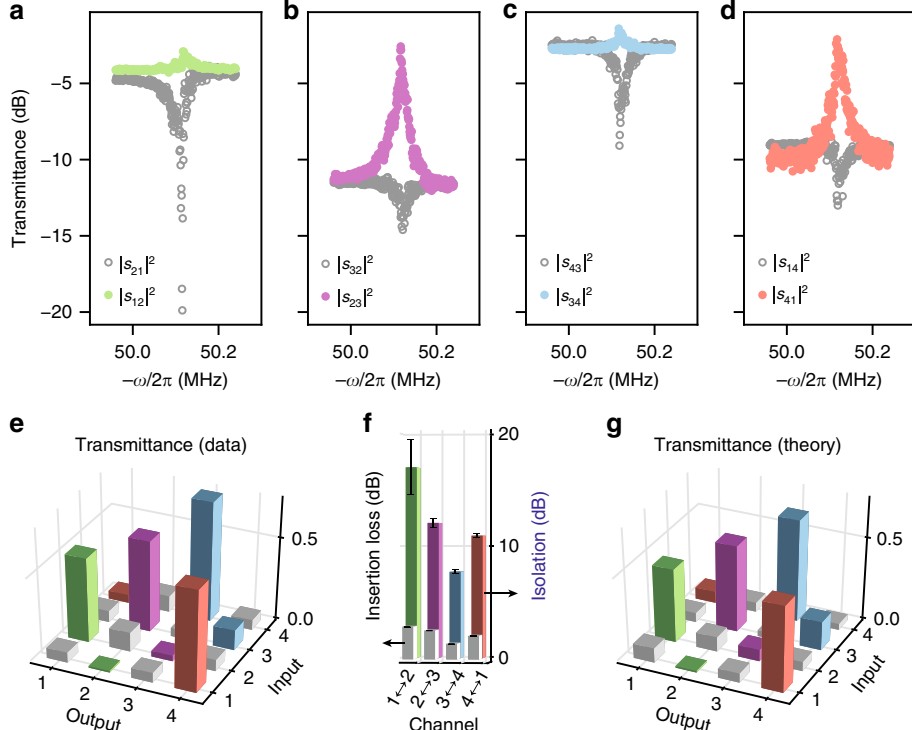

**Fig. 3** Circulation with optomechanical gain. **a–d** Probe transmittances in the four relevant channels, **a** $1 \leftrightarrow 2$, **b** $2 \leftrightarrow 3$, **c** $3 \leftrightarrow 4$, **d** $4 \leftrightarrow 1$, as a function of the control-probe detuning for $\overline{\Delta} = \Omega_m$. The optical splitting results in mechanically induced absorption and transparency for probe beams co-propagating (grey data points) and counter-propagating (coloured data points) to the control beam, respectively. Note that the circulation direction is reversed compared to Fig. 2e. The vertical axis is the same for **a–d**. **e** The $4 \times 4$ transmittance matrix obtained for $\omega_L - \omega_p = \Omega_m$ exhibits a clear asymmetry (see Supplementary Figs. 3 and 4). Opposite matrix elements are the same colour. The reflections and cross-couplings are shown in grey. **f** The corresponding isolation (colours) and insertion losses (grey). The circulator has a maximum isolation of 17 dB with insertion losses <3 dB in all channels. Error bars represent the standard deviation. **g** Predicted $4 \times 4$ transmittance matrix for a blue-detuned control field when $(\kappa_a, \kappa_b)/2\pi = (2.8, 1.8)$ MHz, a mode splitting of $\mu/\pi = 4.3$ MHz (obtained from a global fit, see Supplementary Note 2) and fixed cooperativity of $\mathcal{C} = 0.63$

port circulation with enhanced bandwidth from only two cavity modes and a single external drive field.

**Increasing bandwidth and suppressing backscattering.** The strength of the optomechanical interactions between the optical modes and the mechanical resonance is tuned by the control laser power and can be parametrised by the cooperativities $\mathcal{C}_i \equiv 4|g_i|^2/\kappa\Gamma_m$. Perfect optical circulation, with infinite isolation and zero insertion loss, is approached at high cooperativities and vanishing intrinsic loss $\kappa_0$ (Fig. 1d, e). Stronger nonreciprocity is thus obtained for larger optomechanical coupling than currently achieved in our experiment (Fig. 2, limited by thermal instabilities), which reached $\mathcal{C} \equiv \mathcal{C}_1 + \mathcal{C}_2 = 0.31$ (obtained from a global fit of the $4 \times 4$ transmittance matrix, see Supplementary Note 2). Moreover, the isolation could be increased by increasing the normalised coupling rates $\eta_{a,b} = \kappa_{a,b}/\kappa$ such that the exchange losses to the waveguides overcome the intrinsic cavity losses ($\eta_{a,b} = (0.19, 0.25)$, $\kappa/2\pi = 14.8$ MHz in Fig. 2). Interestingly, as the nonreciprocal transmission in our system is mediated by the optical loss channel, the bandwidth of operation (see Supplementary Note 5) can be tuned to far exceed the mechanical linewidth using high cooperativities and an over-coupled scenario, finally limited only by the mechanical frequency.

Generally, the performance of add-drop filters is limited by a lifting of optical mode degeneracy, quantified by a finite mode splitting $\mu = (\omega_2 - \omega_1)/2$. In ring resonators, such splitting can result from surface inhomogeneities that directly couple clockwise and counterclockwise propagating waves[29]. This changes the relative phase with which both (odd and even) normal modes are

excited, leading to small ($\sim -16$ dB) reflection $\left(|s_{33}|^2\right)$ and cross-coupling $\left(|s_{31}|^2\right)$ signals in our circulator (Fig. 2c, d) indicative of a normalised mode splitting $\delta \equiv 2\mu/\kappa = 0.19$. Interestingly, the optomechanical interaction suppresses these unwanted signals over the nonreciprocal frequency band (Fig. 2c, d). Assuming equal control powers in the optical modes ($g_2 = ig_1$), the resonantly reflected signal at port $j$ of waveguide a, b reads $|s_{jj}|^2 = 4\eta_{a,b}^2 \delta^2 (1 + \mathcal{C} + \delta^2)^{-2}$ (see Supplementary Note 3). A large cooperativity thus annihilates unwanted reflection (and likewise cross-coupling), imprinting an enhanced directionality in the ring resonator that actively mitigates backscattering at imperfections, from which high-$Q$ ring resonators often suffer. This can be understood by recognising that the nonreciprocal mode coupling at rate $\mu_m$ then dominates the direct optical mode coupling $\mu$, making the impact of the latter on optical response negligible.

**Exploiting gain in reconfigurable circulation.** One way to overcome the limitation on isolation due to intrinsic loss is to introduce gain. This naturally occurs in our system when the control beam is detuned to the blue mechanical sideband $\left(\overline{\Delta}_{1,2} \approx \Omega_m\right)$[13]. In this regime, the device still acts as a circulator, but with opposite handedness $1 \longleftarrow 2 \longleftarrow 3 \longleftarrow 4 \longleftarrow 1$, since the mode conversion process for blue-detuned control fields has opposite phase, i.e., $\mu_m = 2g_1 g_2^*/\Gamma_m$. As a result, constructive and destructive intracavity interference happens opposite to the scenarios depicted in Fig. 2h, i. Now, a probe signal co-propagating with the control beam experiences optomechanically induced absorption. Figure 3a–d plots the probe transmittances for the direct and add-drop channels for $\overline{\Delta}_{1,2} \approx \Omega_m$. We indeed observe a

reduction of co-propagating probe signal due to induced absorption (grey data points). Moreover, and resulting from mode splitting, counter-propagating signals (coloured data points) experience OMIT, which significantly enhances the nonreciprocal system response. This response is characterised by inspecting the complete $4 \times 4$ transmittance matrix (for $\omega_L - \omega_p = \Omega_m$) shown in Fig. 3e. Besides displaying a clear asymmetry, we use this matrix to calculate the per-channel isolation ($\sim$10 dB or more) and insertion loss ($\lesssim$3 dB) as displayed in Fig. 3f. The properties of the realised circulator agree well with the prediction of our model for a cooperativity of $\mathcal{C} = 0.63$ (Fig. 3g). Importantly, our theory predicts (see Supplementary Note 3 and Supplementary Fig. 5) that loss and gain can be balanced to give complete blocking (transmittance) in unwanted (preferred) directions. For $\mathcal{C} = 1$, this condition reads $\delta^2 = 2\eta$, resulting in a nonreciprocal bandwidth $\Gamma_m(1 - \mathcal{C}/(1 + \delta^2))$ and finite back-reflections $\delta^2$, which can be relatively small in an undercoupled scenario. The isolation and low insertion loss, combined with the small footprint of our optomechanical circulator, provides an interesting alternative to garnet-based optical circulators for on-chip applications that can tolerate relatively small bandwidths, as garnet-based systems usually feature millimetre size and tens of dBs of insertion loss[5].

## Discussion

In addition to the transport properties, the noise performance of the circulator is important for faithful signal processing, especially for applications in the quantum domain. Under optimal conditions of negligible intrinsic losses and equal losses for the optical modes, the total output noise spectral density, $N_j$, at port $j$ for a red-detuned control is $N_{1,3} = 1/2 + 2\mathcal{C}\delta^2 \overline{n}_{th}/(1 + \mathcal{C} + \delta^2)^2$ and $N_{2,4} = 1/2 + 2\mathcal{C}\overline{n}_{th}/(1 + \mathcal{C} + \delta^2)^2$, where $\overline{n}_{th} = k_B T/\hbar\Omega_m$ is the thermal population of the mechanical resonator (see Supplementary Note 6 for detailed noise analysis). In this regime an increased cooperativity can be beneficial to widen the OMIT feature[28], dampening the mechanical resonator with the optical bath to distribute the thermal noise over a larger bandwidth. In the high-cooperativity limit, the added noise in all transmission channels of the optomechanical circulator vanishes. In our experiments with $T = 296$ K and $\overline{n}_{th} \sim 10^5$, the thermal noise contribution over the detection bandwidth (5 kHz) can be estimated to be $\sim$10 pW, about three orders of magnitude below the signal power of 15 nW on resonance. Nonetheless, high-frequency mechanical resonators exhibiting smaller thermal populations, combined with larger optomechanical cooperativity, would be beneficial to achieve operation in the quantum regime. Interestingly, when working with degenerate optical modes ($\delta = 0$), output ports 1 and 3 do not carry any thermal noise, or additional quantum noise beyond the vacuum for that matter. Active reconfiguration through the optical control field thus presents the possibility of steering noise, choosing to protect for example either the source or receiver of a signal from any added noise.

In contrast, a blue-detuned control reduces the bandwidth of the device and thus requires more stringent conditions on $\overline{n}_{th}$. The optimal condition for circulation in this regime stipulates working in the undercoupled scenario with $\delta^2 = 2\eta$ for a cooperativity of $\mathcal{C} = 1$ (see Supplementary Notes 3 and 4). The added noise in this regime can be shown to be $n_{add,1,3} = 2(\overline{n}_{th} + 1)$ and $n_{add,2,4} = 2(\overline{n}_{th} + 1)/\delta^2$. So, although the thermal contribution to the noise could be negated with sufficient cooling of the mechanical resonator, the added noise is seen to be larger than one and strongly influenced by the tolerance for backreflections and cross-couplings required for the device. As such, to achieve strong circulation at small control powers, blue detuning can be

beneficial, whereas to maximise bandwidth and noise performance, the red-detuned regime is preferred. It would be good to ascertain whether schemes that include more optical or mechanical modes would allow increased flexibility to mitigate noise and improve bandwidth.

In conclusion, we demonstrated nonreciprocal circulation of light through radiation pressure interactions in a three-mode system, with active reconfigurability regulated via the strength, phase and detuning of control fields. Our experiments and theoretical model recognise different regimes of circulating response and reveal the general importance of destructive intracavity interference between direct and mode conversion paths on the device properties. These principles can lead to useful functionality for signal processing and optical routing in compact, on-chip photonic devices.

After completion of this manuscript, related work by Shen et al.[30] was reported in Nature Communications.

## Methods

**Experimental details.** The microtoroid is fabricated using previously reported techniques, see, e.g., ref. [25]. The sample is placed in a vacuum chamber operated at $3 \times 10^{-6}$ mbar and room temperature. Two tapered optical fibres, mounted on translation stages, are positioned near the toroid to couple the waveguides and optical modes. Three electronically controlled $1 \times 2$ optical switches are used in tandem to launch the probe beam into one of the four ports. A schematic of the experimental set-up is provided in Supplementary Note 1 (see Supplementary Fig. 1). A tunable laser (New Focus TLB-6728) at $\omega_L$ is locked to one of the mechanical sidebands of the cavity mode, with a Pound–Drever–Hall scheme, and launched in port 1. The output of a vector network analyser (VNA, R&S ZNB8) at frequency $\omega$ drives a double-parallel Mach–Zehnder interferometer (DPMZI, Thorlabs LN86S-FC) to generate the probe beam at frequency $\omega_p = \omega_L \pm \omega$. The DPMZI is operated in a single sideband suppressed-carrier mode, allowing selection of the relevant sideband at frequency $\omega_p = \omega_L + \omega$ or $\omega_p = \omega_L - \omega$. The power and polarisation of the control and probe beams are controlled by variable optical attenuators and fibre polarisation controllers, respectively. The probe beam exiting a port is combined with the control beam on a fast, low-noise photodetector whose output is analysed by the VNA. All four detectors are connected to the VNA through a $4 \times 1$ radio frequency switch.

**Scattering matrix for the optomechanical circulator.** Considering an optomechanical system that consists of two optical modes coupled to one mechanical mode, the modal amplitudes of the intracavity probe photons $\delta a_i$ ($i = 1, 2$) and the phonon annihilation operator $b$ are governed by the following coupled dynamical equations[13]:

$$\frac{d}{dt}\delta a_i = \left(i\overline{\Delta}_i - \frac{\kappa_i}{2}\right)\delta a_i + ig_i\left(b + b^\dagger\right) + \sum_{j=1}^{4} d_{ji}\delta s_j^+ \qquad (1)$$

$$\frac{d}{dt}b = \left(-i\Omega_m - \frac{\Gamma_m}{2}\right)b + i\sum_{i=1}^{2} g_i^* \delta a_i + g_i \delta a_i^\dagger \qquad (2)$$

In these relations, the loss rate $\kappa_i$ of each optical mode $i$ is defined as $\kappa_i \equiv \kappa_{0,i} + \kappa_{a,i} + \kappa_{b,i}$, where $\kappa_{0,i}$ refers to the intrinsic optical loss rate of the mode, and $\kappa_{a,i}$, $\kappa_{b,i}$ are the exchange losses to waveguides a and b. The D-matrix elements $d_{ji}$ specify the coupling between input fields $\delta s_j^+$, incident from port $j$, and the optical modes. In addition, $g_{1,2} = g_0 \alpha_{1,2}$ and $\overline{\Delta}_{1,2} = \Delta_{1,2} + (2g_0^2/\Omega_m)\left(|\alpha_1|^2 + |\alpha_2|^2\right)$, respectively, represent the enhanced optomechanical coupling and the modified frequency detunings, where $\Delta_{1,2} = \omega_L - \omega_{1,2}$ is the frequency detuning of the control laser with respect to the optical resonance frequencies. Here, we consider a frequency splitting of $2\mu$ between the resonance frequencies of the two optical modes, and without loss of generality assume $\omega_{1,2} = \omega_0 \mp \mu$, such that $\overline{\Delta}_{1,2} = \overline{\Delta} \pm \mu$. Equations (1) and (2) completely describe the behaviour of the system in connection with the input–output relations:

$$\begin{pmatrix} s_1^- \\ s_2^- \\ s_3^- \\ s_4^- \end{pmatrix} = C \begin{pmatrix} s_1^+ \\ s_2^+ \\ s_3^+ \\ s_4^+ \end{pmatrix} + D \begin{pmatrix} a_1 \\ a_2 \end{pmatrix} \qquad (3)$$

where the matrix $C$ defines the port-to-port direct-path scattering matrix of the optical circuit, while $D$ and its transpose $D^T$ describe the mode-to-port and port-to-mode coupling processes, respectively. Using the rotating wave approximation,

Eqs. (1), (2) and (3) lead to the frequency-domain scattering matrix:

$$S = C + iD(M + \omega I)^{-1}D^{\mathrm{T}}$$

$$S = C + iD \begin{pmatrix} \Sigma_{o_1} \mp |g_1|^2/\Sigma_m^{\pm} & \mp(g_1 g_2^*)/\Sigma_m^{\pm} \\ \mp(g_1^* g_2)/\Sigma_m^{\pm} & \Sigma_{o_2} \mp |g_2|^2/\Sigma_m^{\pm} \end{pmatrix}^{-1} D^{\mathrm{T}}. \quad (4)$$

Here the upper and lower signs correspond to a control laser that is red-$(\overline{\Delta} = -\Omega_m)$ and blue-detuned $(\overline{\Delta} = +\Omega_m)$ with respect to the optical resonance. In addition, $\Sigma_m^{\pm} \equiv \omega \mp \Omega_m + i\Gamma_m/2$ and $\Sigma_{o_{1,2}} \equiv \omega + \overline{\Delta}_{1,2} + i\kappa_{1,2}/2$, respectively, represent the inverse mechanical and optical susceptibilities, where $\overline{\Delta}_{1,2} \equiv \overline{\Delta} \pm \mu$. Note that $\mu_m$, the mechanically mediated mode conversion rate, is identified from the off-diagonal elements of the $M$ matrix.

The direct-path scattering matrix $C$ for the side-coupled geometry that describes both systems in Fig. 1a, c is

$$C = \begin{pmatrix} 0 & 1 & 0 & 0 \\ 1 & 0 & 0 & 0 \\ 0 & 0 & 0 & 1 \\ 0 & 0 & 1 & 0 \end{pmatrix}. \quad (5)$$

The $D$ matrix is obtained from symmetry considerations, power conservation $(D^{\dagger}D = \mathrm{diag}(\kappa_{a,1} + \kappa_{b,1}, \kappa_{a,2} + \kappa_{b,2}))$ and time-reversal symmetry $(CD^* = -D)$[31]. For the even and odd mode picture, the $D$ matrix reads

$$D = \frac{1}{\sqrt{2}} \begin{pmatrix} i\sqrt{\kappa_{a,1}} & -\sqrt{\kappa_{a,2}} \\ i\sqrt{\kappa_{a,1}} & \sqrt{\kappa_{a,2}} \\ i\sqrt{\kappa_{b,1}} & -\sqrt{\kappa_{b,2}} \\ i\sqrt{\kappa_{b,1}} & \sqrt{\kappa_{b,2}} \end{pmatrix}. \quad (6)$$

Equation (4), combined with Eqs. (5) and (6) fully specifies the scattering matrix of our four-port circulator. We assume equal losses $(\kappa_{0,1} = \kappa_{0,2} = \kappa_0, \kappa_{a,1} = \kappa_{a,2} = \kappa_a, \kappa_{b,1} = \kappa_{b,2} = \kappa_b)$ and equal control powers $(g_2 = ig_1 = ig)$ in the two optical modes to fit the experimentally measured transmittances $(|s_{ij}|^2)$ using a global fitting procedure (see Supplementary Note 2).

**Data availability**. The data that support the findings of this study are available from the corresponding author on reasonable request.

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

## Acknowledgements

This work has been supported by the Office of Naval Research, with grants no. N00014-16-1-2466 and N00014-15-1-2685. It is part of the research programme of the Netherlands Organisation for Scientific Research (NWO). E.V. acknowledges support by the European Union's Horizon 2020 research and innovation programme under grant agreement no. 732894 (FET Proactive HOT).

## Author contributions

F.R. developed the experimental set-up. F.R. and J.P.M. performed the experiments and analysed the data. M.-A.M. developed the theoretical model, with contributions from E.V., A.A., F.R, and J.P.M. E.V. and A.A. supervised the project. All authors contributed to the writing of the manuscript.
