## [Peer Review File · Nature Communications]

Reviewers' comments:

Reviewer #1 (Remarks to the Author):

The paper by Ruesink et al presents a demonstration of an optical circulator based on an optomechanical interaction which mimicks the presence of an effective magnetic field breaking reciprocity. The device works at room temperature and at telecommunication wavelength providing in this way an important proof-of-principle of an optical circulator which could be integrated in a circuit and does not require bulky magnetic materials. The paper is very well written, it clearly provides evidence of the results and all the experimental details are provided in a clear way in the method section and in the supplementary material. I have found no weak point in the paper and in the presentation, and due to the timeliness of the topic (it is the first clear demonstration of a non-magnetic circulator at optical wavelength) I suggest its publication in Nature Communication in the present form.

Reviewer #2 (Remarks to the Author):

In the manuscript, the authors demonstrate the implementation of an optical circulator based on the coupling of two optical fibers to a silica microtoroid. By using the broken symmetry induced by an external drive, they enable a non-reciprocal connectivity between the four ports with decent isolation and insertion loss that is not too bad. Doing so, they join the now growing literature of implementing non-reciprocal devices using external driving in optomechanical, electromechanical, and Josephson systems. The results are solid and I see no technical faults in the manuscript.

On the other hand, I find that the results themselves are not so novel: the idea and implementation of non-reciprocity were already demonstrated by the authors in a previous publication in Nature Communications, in which they demonstrated an optical isolator. In this work, they add an additional optical fiber coupled to the microtoroid and extend this approach to include circulation in addition to the previously demonstrated isolation. In this sense, I am not convinced that on this basis the result is suitable for this journal.

Of course, a circulator itself has potentially different applications as compared to an isolator. To make a case for this, I miss some of the practical selling points of this implementation: how does this compare to state-of-the-art optical circulators? Does it offer comparable bandwidth? What is the added noise in this circulator? How does this compare to current commercial optical circulators?

What is the relevance of this device for non-quantum applications: would it be relevant for replacing current optical circulators in commercial applications? Are the dynamic range / power handling capabilities of the device?

If the current device far exceeds the performance of current commercial circulators, I could imagine this could have potential impact. However, my impression is that this is a relatively straightforward extension of the already demonstrated proof-of-principle device by the same authors, and therefore find it is more suitable for a journal such as Applied Physics Letters.

Response to reviewers' comments on "Optical circulation in a multimode optomechanical resonator" by F. Ruesink et al.

Reviewer #1 (Remarks to the Author):

The paper by Ruesink et al presents a demonstration of an optical circulator based on an optomechanical interaction which mimicks the presence of an effective magnetic field breaking reciprocity. The device works at room temperature and at telecommunication wavelength providing in this way an important proof-of-principle of an optical circulator which could be integrated in a circuit and does not require bulky magnetic materials. The paper is very well written, it clearly provides evidence of the results and all the experimental details are provided in a clear way in the method section and in the supplementary material. I have found no weak point in the paper and in the presentation, and due to the timeliness of the topic (it is the first clear demonstration of a non-magnetic circulator at optical wavelength) I suggest its publication in Nature Communication in the present form.

Response to reviewer #1

We thank the reviewer for his/her thorough reading of our manuscript and for the recommendation of 'publication in Nature Communications in the present form'. We are pleased to know that the reviewer finds no weak point in the paper and recognizes that our work is novel while presenting a significant advancement to the fields of optomechanics and photonics.

Reviewer #2 (Remarks to the Author):

In the manuscript, the authors demonstrate the implementation of an optical circulator based on the coupling of two optical fibers to a silica microtoroid. By using the broken symmetry induced by an external drive, they enable a non-reciprocal connectivity between the four ports with decent isolation and insertion loss that is not too bad. Doing so, they join the now growing literature of implementing non-reciprocal devices using external driving in optomechanical, electromechanical, and Josephson systems. The results are solid and I see no technical faults in the manuscript.

On the other hand, I find that the results themselves are not so novel: the idea and implementation of non-reciprocity were already demonstrated by the authors in a previous publication in Nature Communications, in which they demonstrated an optical isolator. In this work, they add an additional optical fiber coupled to the microtoroid and extend this approach to include circulation in addition to the previously demonstrated isolation. In this sense, I am not convinced that on this basis the result is suitable for this journal.

Response to reviewer #2

We thank the reviewer for his/her detailed analysis of our manuscript and providing pointers for improving it. We are pleased to know that the reviewer finds the results of our proof-of-principle optical circulator to be of good quality and relevant to the wider opto- and electromechanics communities.

We firmly believe that our manuscript provides important novelty. This novelty not only lies in the fact that it presents the 'first clear demonstration of a non-magnetic circulator at optical wavelength' (to use the words of reviewer #1) but also in the fact that it is not at all generally straightforward to turn an optomechanical isolator – or any isolator based on parametric modulation – into a circulator. We identify the microtoroid ring resonator coupled to two waveguides as one experimental implementation of a general class of minimal optomechanical circulators, and explain the crucial requirements on mode coupling and interference in the output ports that allow circulation, with potentially ideal efficiency and

favorable bandwidth properties. We recognize from the reviewer's comments that we had not clearly enough stressed these points in our original manuscript. Below, and in the revised manuscript, we explain them in more detail.

We point out that in general, for multimode optomechanical systems that achieve two-port isolation, the addition of one or two extra ports does not directly result in circulation. Many of the opto- and electromechanical isolators demonstrated recently use the mechanical bath to dissipate energy in the reverse direction [1-4]. The bandwidth of such systems therefore typically does not exceed the mechanical linewidth [5]. Adding an additional waveguide port does not replace that bath, and as such is insufficient to create circulation. Ideas to nonetheless use such systems to create circulators need at least 3 optical cavity modes controlled by as many as 6 tuned drive fields [1, 4].

We find that circulation can be realized in any two-mode optical system if its couplings to the ports are designed such that light is transmitted from one port to a second when not interacting with the cavity, and transmitted to a third (via the cavities) when on resonance. This is explained in Fig. 2, which shows the behavior of a general two-mode system that is entirely equivalent to the ring resonator implementation we use. Then, only a single input drive is needed to create the proper nonreciprocal transport. Moreover, the bandwidth in this scheme is enhanced by the drive field, in principle far beyond the mechanical linewidth for high cooperativities. And, as we show in Fig. 3, we also identify a new regime where near-ideal circulation is achieved at moderate cooperativity for narrow-bandwidth operation, by balancing loss and optomechanical gain. By describing the requirements to achieve such operation for general multimode systems, we point the way to implement it in different opto- or electromechanical systems (including superconducting circuits, photonic crystals, etc.), different frequency regimes, and potentially even in systems that use different time-modulation mechanisms.

Action taken:

We changed the manuscript to highlight the importance of the port coupling conditions in allowing us to demonstrate circulation, as well as the breadth of its implications. Main change is the addition of a paragraph below Figure 2 that explains the novelty of our work in this context. Moreover, we explicitly mention the significance of our demonstration of on-chip magnet-free circulation at optical wavelengths.

Reviewer #2 (Remarks to the Author, continued):

Of course, a circulator itself has potentially different applications as compared to an isolator. To make a case for this, I miss some of the practical selling points of this implementation: how does this compare to state-of-the-art optical circulators? Does it offer comparable bandwidth? What is the added noise in this circulator? How does this compare to current commercial optical circulators? What is the relevance of this device for non-quantum applications: would it be relevant for replacing current optical circulators in commercial applications? Are the dynamic range / power handling capabilities of the device? If the current device far exceeds the performance of current commercial circulators, I could imagine this could have potential impact. However, my impression is that this is a relatively straightforward extension of the already demonstrated proof-of-principle device by the same authors, and therefore find it is more suitable for a journal such as Applied Physics Letters.

Response to reviewer #2

The reviewer asks for a comparison of our device to state-of-the-art optical circulators. The technological challenge that we are contributing to is the creation of compact, on-chip photonic circulators – which are currently not commercially available, to the best of our knowledge. The best comparison we can make is

to reported on-chip alternatives that rely on magneto-optic effects using garnets. Such devices rely on the magneto-optic effect in garnet crystals that are deposited or bonded onto the chip surface. In these devices, the phase differences necessary for non-reciprocal operation is achieved over length scales larger than $300\ \mu\text{m}$. This, combined with the size of the garnet crystal, restricts the overall device length to $\sim 1\ \text{mm}$. The presence of a large garnet crystal over the on-chip waveguides presents the additional disadvantage of insertion loss. The insertion loss in these devices is in practice 10's of dBs of which the loss due to the active part of the device alone is estimated to be 10-13 dB. In fact, the insertion loss is fundamentally limited by the absorption of the garnet crystal to more than 3 dB [6].

On the other hand, optomechanics-based non-reciprocal devices, like the one demonstrated in our manuscript, have a footprint that is at least two orders of magnitude smaller. Further, the insertion loss in our device is much smaller ($< 3\ \text{dB}$) and limited only by experimental challenges and not fundamental limitations. To the best of our knowledge, this is the first demonstration of a microscale, low-loss optical circulator of any kind. For larger optomechanical cooperativities the insertion loss can be made vanishingly small as seen from the theoretical plots in Fig. 1 of the manuscript. The combination of smaller scale and lower insertion loss allows our devices to be efficiently integrated into on-chip photonic circuits. That said, it is important to acknowledge also the limitations of our device, as we do in our manuscript: With a single mechanical mode, the bandwidth is ultimately bounded by the mechanical frequency. The device adds noise in some ports due to upconversion of thermomechanical fluctuations; in our manuscript, we quantitatively discuss how this is reduced (potentially below the quantum level) through optomechanical cooperativity. Finally, the dynamic range is limited by the pump power – but the range of linear operation is much larger than for example in Josephson-based devices.

An interesting added advantage in our device is the reconfigurability offered by the choice of the control detuning and direction. The ability to actively control the direction of circulation, and hence the direction of signal and noise propagation, could be a useful feature for on-chip applications. In addition, the role of the optical loss channel in achieving non-reciprocal transmission provides the added functionality of a tunable bandwidth.

Finally, we would like to remark, with all due respect, that we do not agree with the notion that this work has “potential impact” only if our device “far exceeds the performance of current commercial circulators.” We do not believe that all new science is void of impact if it does not immediately present performance beyond commercial technology. We hope that we have explained clearly enough where the impact of our work lies.

Action taken:

We added text to the paragraph before the Discussion section to clarify the advantages of our optical circulator over garnet-based on-chip optical circulators.

We hope that we have answered both concerns of the reviewer and that with the added clarifying text our manuscript is deemed suitable for publication in Nature Communications.

[1] Bernier, N. R. et al. Nonreciprocal reconfigurable microwave optomechanical circuit. *Nat. Commun.* **8**, 604 (2017).

[2] Peterson, G. A. et al. Demonstration of efficient nonreciprocity in a microwave optomechanical circuit. *Phys. Rev. X* **7**, 031001 (2017).

[3] Fang, K. et al. Generalized non-reciprocity in an optomechanical circuit via synthetic magnetism and reservoir engineering. *Nat. Phys.* **13**, 465{471 (2017).

- [4] Barzanjeh, S. et al. Mechanical on-chip microwave circulator. *Nat. Commun.* **8**, 953 (2017).
- [5] Miri, M.-A., Ruesink, F., Verhagen, E. & Alù, A. Optical nonreciprocity based on optomechanical coupling. *Phys. Rev. Appl.* **7**, 064014 (2017).
- [6] Shoji, Y. & Mizumoto, T. Magneto-optical non-reciprocal devices in silicon photonics. *Sci. Technol. Adv. Mater.* **15**, 014602 (2014).

REVIEWERS' COMMENTS:

Reviewer #2 (Remarks to the Author):

I would like to thank the authors for their patience and care in replying to my comments.

I now understand more clearly the work, and in particular the relation to earlier works, the novelty, and the relevance. I also thank the authors for incorporating changes to the manuscript that also makes this clear for other potential readers.

For me, particularly the explanation of the criteria for circulation with four ports and the relations to other proposals to achieve similar behaviour with significantly more complicated configurations, makes it more clear to me the novelty of the work.

On another point: I also agree with the authors completely that work that is based on pure science has absolute value to the community, and should absolutely be published, even if devices developed do not exceed the state of the art. My comment was more with regards my perception of the criteria for impact for Nature Communications, which I am not sure would have been met without the novelty identified by the authors in their rebuttal, and reflected in the changes in clarity in the manuscript.

Considering the above discussion, I am now able to recommend the manuscript for publication in Nature Communications.

Response to the reviewer's comments on "Optical circulation in a multimode optomechanical resonator" by F. Ruesink et al.

Reviewer #2 (Remarks to the Author):

I would like to thank the authors for their patience and care in replying to my comments.

I now understand more clearly the work, and in particular the relation to earlier works, the novelty, and the relevance. I also thank the authors for incorporating changes to the manuscript that also makes this clear for other potential readers.

For me, particularly the explanation of the criteria for circulation with four ports and the relations to other proposals to achieve similar behaviour with significantly more complicated configurations, makes it more clear to me the novelty of the work.

On another point: I also agree with the authors completely that work that is based on pure science has absolute value to the community, and should absolutely be published, even if devices developed do not exceed the state of the art. My comment was more with regards my perception of the criteria for impact for Nature Communications, which I am not sure would have been met without the novelty identified by the authors in their rebuttal, and reflected in the changes in clarity in the manuscript.

Considering the above discussion, I am now able to recommend the manuscript for publication in Nature Communications.

Response to reviewer #2

We would like to thank the reviewer for his/her reading of our rebuttal and modified manuscript and recommending publication in Nature Communications. We are pleased to know that the reviewer now fully appreciates the novelty and relevance of our work. We also appreciate the reviewer for explaining his/her view on the merit of fundamental research in comparison to state-of-the-art devices, which concurs with our own.